# An Experimental Study on Dynamic Characteristics of Coarse-Grained Soil under Step Cyclic Loading

Peisen Wang [1], Wenjun Hu [2,*], Pingyun Liu [2], Zhenqiang Yan [3], Xianghui Kong [2,*], Quanman Zhao [2] and Wenhao Yin [2]

1   School of Civil Engineering, Shandong Jianzhu University, Jinan 250101, China; wangpeisen@sdjzu.edu.cn
2   School of Transportation Engineering, Shandong Jianzhu University, Jinan 250101, China; lpy1802@163.com (P.L.); zhaoquanman@sdjzu.edu.cn (Q.Z.); ywh456321@163.com (W.Y.)
3   Jinan Jinqu Road Survey Design Research Co., Ltd., Jinan 250101, China; fangzhang_july@163.com
*   Correspondence: huwenjun@sdjzu.edu.cn or huwenjun_82@163.com (W.H.); kongxh@sdjzu.edu.cn (X.K.)

**Abstract:** The accumulated plastic deformation induced by a cyclic traffic load will lead to destruction of the subgrade. Coarse-grained soil is a widely used subgrade filler. The GDS dynamic triaxial test was carried out on typical coarse-grained soil fillers to investigate the influence of different confining pressures, consolidation ratios and numbers of cyclic actions on the hysteresis curve under step cyclic loading. The results show that surrounding pressure can significantly reduce the energy lost from a soil sample under cyclic loading. Under the same stress level, increasing the consolidation ratio can effectively reduce the area enclosed by the hysteresis curve. When the stress is increased above a certain value, the strain of the response clearly changes with an increase in the number of cyclic loading. The research findings can theoretically guide the design of coarse-grained soil roadbeds in practical engineering.

**Keywords:** subgrade engineering; coarse-grained soil; step cyclic loading; dynamic triaxial test; hysteresis curve



## 1. Introduction

Research on the dynamic properties of coarse-grained soils mainly focuses on the dynamic deformation problems [1–3] caused by road traffic loads, which have resulted in a large number of disease phenomena. Normally, traffic loading on roads is found to be transient, uncertain and repetitive [4]. Thus, it is difficult to formulate traffic loads. The dynamic stress–strain relationship provides an intuitive understanding of the strain development pattern of coarse-grained soils under dynamic loading. Long Y. et al. [5] introduced the effects of the surrounding pressure and cyclic stress ratio on the accumulated plastic strain of coarse-grained soil into the hyperbolic prediction model showing proper prediction results, which consider the influence of cyclic stress, static deviational stress and consolidation pressure. Chen Y.P. et al. [6] proposed an improved exponential model to describe the development law of cumulative plastic strain before the soil was destroyed. Tang et al. [7] introduced the concept of equivalent vibration times and proposed a model for predicting the cumulative plastic strain index in soils under multi-stage dynamic stress loading conditions. Bian [8] investigated the effects of the circumferential pressure and dynamic amplitude on the properties of ballast, such as axial strain and volumetric strain. Trinh et al. [9] investigated the effects of the water content and saturation on the mechanical properties of coarse-grained soil specimens in contaminated ballast layers under old railroad structures and developed a principal structure model, which considers the stress level, number of cycles and water content of the soil. Leng W.M et al. [10] studied the effects of the dynamic stress amplitude, water content and surrounding pressure on the cumulative deformation pattern of coarse-grained soils and proposed the stability limits and discrimination criteria for dynamic deformation of coarse-grained soils. In summary,

investigations on the dynamic properties of coarse-grained soils under step cyclic loading through experimental methods are limited. In this paper, the GDS dynamic triaxial test was carried out to investigate the effect of different confining pressures, consolidation ratios on the hysteresis curve of typical coarse-grained soil roadbed under step cyclic loading.

In conclusion, it was necessary to study the dynamic characteristics of coarse-grained soil under step cyclic loading by experimental methods. The GDS dynamic triaxial system is an indoor geotechnical triaxial test instrument controlled by a motor and produced by GDS instrument and Equipment Co., Ltd. (Hook Hampshire, UK) The dynamic triaxial test belonged to the dynamic test range of soil. It is a commonly used method to study the dynamic characteristics of soil in a laboratory. The dynamic triaxial apparatus is a commonly used indoor geotechnical test device. In order to understand the dynamic characteristics of coarse-grained soil, this indoor dynamic triaxial test of coarse-grained soil was carried out, which considers the key influencing factors of a soil dynamic triaxial test: the confining pressure, consolidation ratio and so on. The influence of different confining pressures and consolidation ratios on the hysteretic curve of the soil samples under step cyclic loading was explored.

## 2. Test Preparation

### 2.1. Raw Materials

The soil sample used in the test was taken from coarse-grained soil along a new highway, and the granulometry test was performed according to the Test Methods of Soils for Highway Engineering (JTG 3430-2020) [11], which is shown in Table 1. The particle size accumulation curve is shown in Figure 1.

**Table 1.** Particle analysis results.

| Mesh Size (mm) | 10 | 5 | 2 | 1 | 0.5 | 0.25 | 0.1 | 0.075 |
|---|---|---|---|---|---|---|---|---|
| Pass quality percentage (%) | 100 | 95.3 | 74.8 | 62.1 | 50 | 33.8 | 19.5 | 8.3 |

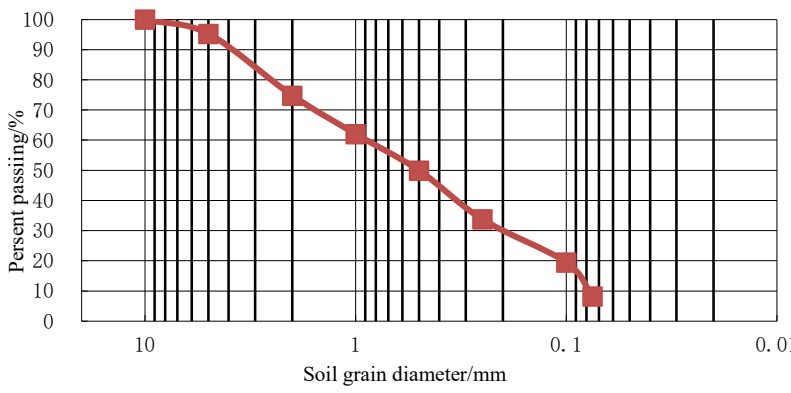

**Figure 1.** Cumulative particle size curve of soil samples.

The mass percentage of soil particles with a diameter above 0.075 mm ($d \geq 0.075$ mm) in the coarse-grained group of soil samples is 91.7%, which was greater than 50% of the total mass. The sample soil is classified as a coarse-grained soil, according to Chinese standard JTG 3430-2020 [11], whose basic physical and mechanical property indexes are shown in Table 2. It can be seen that the particle distribution of this coarse-grained soil was good.

Table 2. Basic performance indicators of soil samples.

| Natural Water Content (%) | Maximum Dry Density /(g/cm³) | Optimum Moisture Content (%) | CBR (%) | Nonuniformity Coefficient $C_U$ (>5) | Curvature Factor $C_C$ (1~3) | Gradation |
|---|---|---|---|---|---|---|
| 2.1 | 2.13 | 8.03 | 51 | 5.6 | 1.1 | well |

The nonuniformity coefficient $C_U$ is calculated with the following equation:

$$C_U = \frac{d_{60}}{d_{10}} \tag{1}$$

The curvature factor $C_C$ is calculated with the following equation:

$$C_C = d_{30}^2 / (d_{60} \times d_{10}) \tag{2}$$

where $d_{60}$, $d_{30}$ and $d_{10}$ = the cumulative percentage of soil weighing less than a certain particle size: 60%, 30% and 10%, respectively.

*2.2. Test Principle*

The dynamic triaxial test was used to analyze the dynamic response of the specimen under dynamic loading by applying a periodic axial dynamic principal stress to the specimen on the basis of the applied axial static stress. Based on the relative relationship of dynamic indicators, such as stress, strain and pore pressure, the dynamic properties [12–19] of the soil and the properties of the soil specimen under dynamic stress are derived. Stress is usually expressed in terms of axial major principal stress $\sigma_1$ and circumferential pressure $\sigma_3$, which refer to the stress state of the soil under static and dynamic conditions. The dynamic conditions are various parameters of the simulated dynamic load, mainly including its vibration direction, waveform, frequency, amplitude, etc. The test uses the GDS vibration triaxial instrument (GDS Instruments, Hook Hampshire, UK), which consists of a triaxial pressure chamber, an axial and lateral pressurization system, a counterpressure pressure volume controller, a data collector and a computer, which can accurately complete the small strain triaxial test under dynamic stress.

Referring to the experience of previous soil dynamic triaxial research, the failure standard of the specimen was 5% of the axial dynamic strain. Considering that some specimens were not damaged, we took 25,000 vibrations as the second test termination standard. The dynamic loading frequency range of the GDS dynamic triaxial apparatus used in the test was 0–2 Hz; the dynamic axial pressure could be loaded ± 10 kN, and the accuracy was 0.1% of the full scale. The displacement range was 100 mm, the displacement resolution was 0.208 mm, and the axial displacement accuracy was 0.07% of the full range. The above ranges met the test requirements. The GDS dynamic triaxial apparatus is a fully automatic dynamic triaxial apparatus. It can realize the saturation, detection, consolidation and dynamic loading of a sample. Except for installation of a sample, the other operations were completely controlled by the computer, which accurately completed each step of operation, effectively reduced human errors and ensured smooth progress of the test. The complete device is shown in Figure 2.



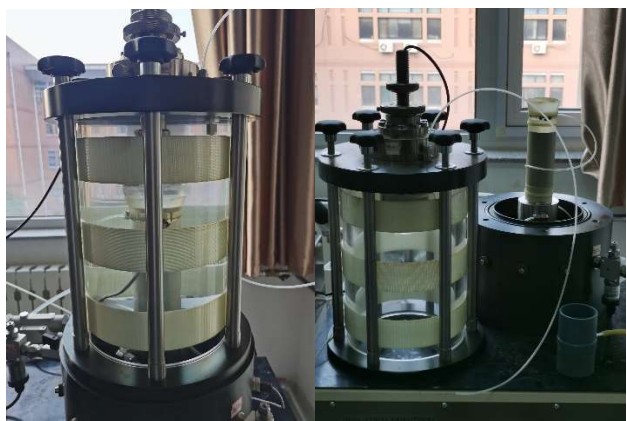

**Figure 2.** The complete device.

*2.3. Test Program*

In order to simulate the stress state of the soil sample in the actual situation more accurately, we selected four aspects of the sample parameters: soil conditions, drainage conditions, stress conditions and dynamic conditions. We also chose the main influencing factors, including the confining pressure, dynamic stress amplitude, loading frequency [20] and number of cycles. The specific test scheme is shown in Table 3.

(1)  Specimen specification: $\varphi$ 50 mm $\times$ h 100 mm.
(2)  Moisture content of the specimen: We adopted the optimum moisture content, and the standard compaction test result was 8%.
(3)  Specimen compaction: We adopted compaction of 100%, divided into three layers and tamped manually.
(4)  Loading method: We adopted the loading method step-by-step. The dynamic stress amplitude started from 20 kPa, and then increased by 20 kPa at each stage of loading. Loading was stopped when the dynamic stress amplitude reached 200 kPa or the cumulative deformation reached 5% of the initial height. The number of cycles during each stage of loading was 20.
(5)  Drainage conditions: without drainage.

**Table 3.** Dynamic triaxial test scheme.

| Surrounding Pressure (KPa) | Dynamic Stress Amplitude (KPa) | Solidification Ratio | Control Mode | Frequency (Hz) |
|---|---|---|---|---|
| 50 | 20 40 — — — — — | | | |
| 100 | 20 40 60 80 100 — — | 1/1.5 | Stress control | 1 |
| 150 | 20 40 60 80 100 120 140 | | | |

The entire research process is shown in Figure 3.

*2.4. Test Steps*

(1)  Triaxial sample preparation: We prepared the crushed coarse-grained soil sample after drying at the optimum moisture content, mixed them evenly after preparation, and sealed and stored them for 24 h, to make the soil sample fully wet.
(2)  Sample saturation: We adopted vacuum saturation, as shown in Figure 4. After completing vacuum saturation, we tested the B value of the sample. After the B value reached 0.95 or above, the sample was saturated.
(3)  Sample installation: We adjusted the base to proper position, by exhausting the base to ensure that the center axis of the sample coincides with the center of the upper

fixed axis. Then, we installed the pressure chamber and, finally, filled the pressure chamber with water.

(4) Sample consolidation: After saturation, we set the consolidation pressure and consolidated the sample. When the volume rate of the back-pressure drainage was lower than 5 mm$^3$/5 min, the consolidation of the sample was considered completed.

(5) Applying dynamic load: According to the requirements of the test scheme, we gradually applied the load to the sample.

(6) Unloading and removing sample: After completing loading, we reduced the confining pressure and back pressure to 0, drained the water, removed the samples and cleaned the instrument.

Sample preparation: Preparing according to the best moisture content, mixing evenly after preparation, sealing and storing for 24 h to make the soil sample fully wet.

Sample saturation: vacuum saturation

Sample test: Testing the B value of the sample. B = (pore pressure increase value / confining pressure increase value). If the B value reached 0.95 or above, the sample was saturated.

Sample consolidation: After saturation, setting the consolidation pressure and consolidate the sample. When the volume rate of back pressure drainage was lower than 5 mm$^3$ / 5 min, the consolidation of the sample was considered to be completed.

**Figure 3.** The entire research process.

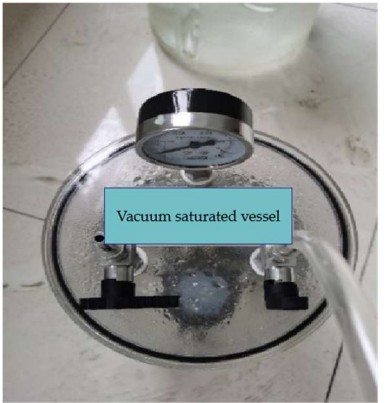 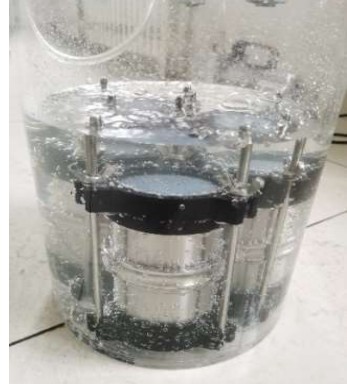 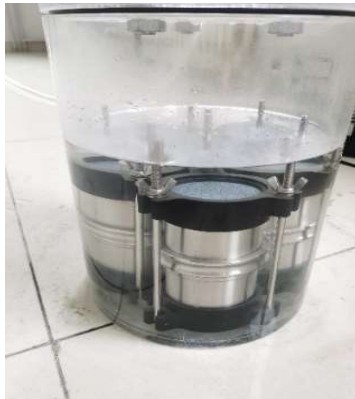

(**a**) Vacuum saturated containers (**b**) During vacuum saturation (**c**) End of vacuum saturation

**Figure 4.** Vacuum saturation.

The adopted breakdown criteria were:

(1) When the sample was subjected to a cyclic dynamic load, the increase in pore water pressure reached the initial consolidation confining pressure for the first time. At this time, the effective stress was zero or minimum, and the test was terminated.

(2) When the specimen was subjected to a cyclic dynamic load and the double amplitude axial strain reached 5%, the test was terminated.

In our experiments, (1) was selected as the failure standard of the soil liquefaction resistance strength, and (2) was selected as the failure standard of the soil dynamic strength.

The breakdown samples are shown in Figure 5.

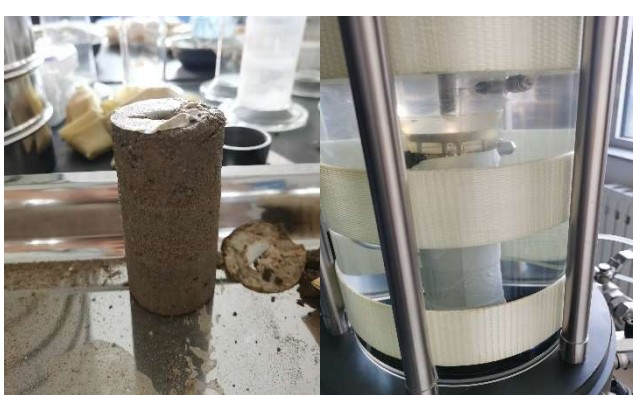

**Figure 5.** Breakdown samples.

## 3. Analysis of Test Results for Hysteretic Curve of Coarse-Grained Soil

When cyclic stress was applied to the soil, the corresponding response strain also performed cyclically. Through the characteristics of the hysteretic curve, the deformation characteristics and energy consumption characteristics of soil under cyclic stress can be learned.

### 3.1. Analysis of Hysteretic Curve Characteristics under Different Confining Pressures

Figure 6 shows the influence of different confining pressures on hysteretic curves under isobaric consolidation. It can be intuitively found that the confining pressure of different sizes has a great influence on the hysteretic curve. When the dynamic stress amplitude was 40 kPa, and the confining pressure was 50, 100 and 150 kPa, the maximum deformation of the sample was 0.275%, 0.061% and 0.057%, respectively. The increase in confining pressure induces a denser internal spatial structure of the soil. Furthermore, when the stress is constant, the increase in confining pressure reduces the strain of the specimen, which shows that increasing the confining pressure can effectively inhibit the development of soil deformation.

In Figure 7, it can be seen that the confining pressure has a great impact on the surrounding area of the hysteretic curve. With the increase in confining pressure, the surrounding area of the hysteretic curve showed the characteristics of a significant decrease. The confining pressure increased from 50 to 150 kPa under a dynamic stress amplitude of 40 kPa. In the isobaric consolidation, the area enclosed by the hysteretic curve decreases by 86% and 93%; In the bias consolidation, the surrounding area of hysteretic curve decreased by 94%. The confining pressure has a great influence on the dissipated energy of coarse-grained soil specimens. Increasing the confining pressure can significantly reduce the energy consumption under cyclic stress. The stiffness and overall stability of the coarse-grained soil specimens improved, which demonstrates improved strength characteristics.

Figure 8 shows the second-degree parabola fitting relationship of the surrounding area of the hysteretic curve under different dynamic stress amplitudes. The correlation coefficients were 0.99 and above, indicating that the second-degree parabola fitting has a high correlation. In isobaric consolidation, under the same dynamic stress amplitude, the slope of the parabola–tangent fitting decreased with an increase in the confining pressure, and the bias consolidation also had the same trend. It showed that under the same stress level, increasing the confining pressure can effectively reduce the area surrounded by the hysteretic curve. An increase in the confining pressure can effectively inhibit the rate of increase in its area with the increase in stress, thus controlling its energy consumption at a high stress level and showing better overall stability.

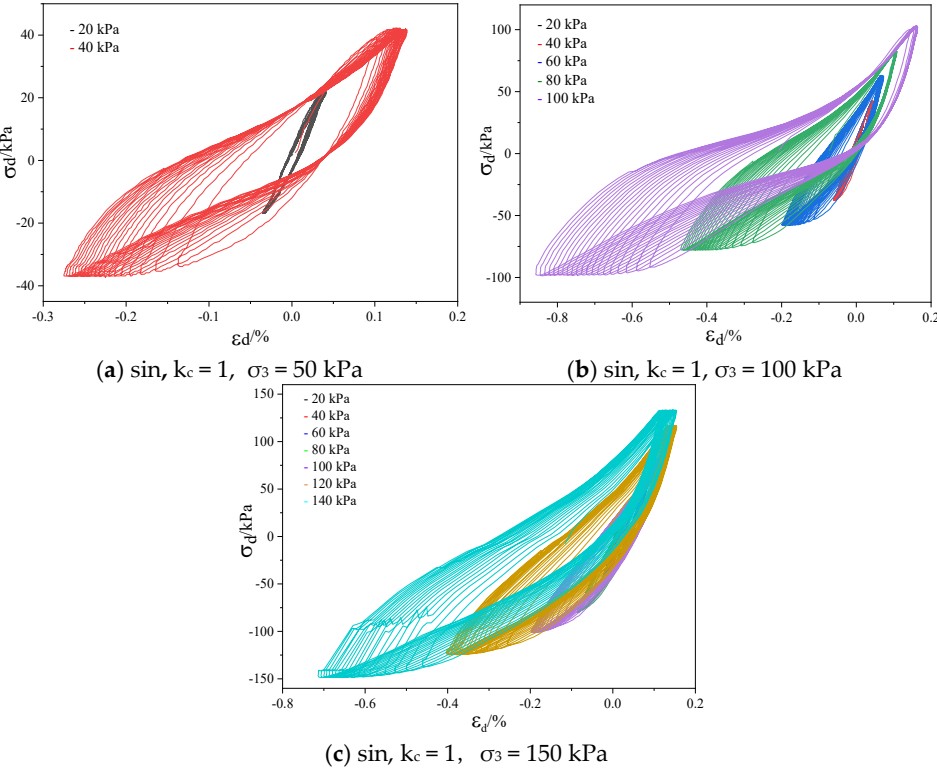

**Figure 6.** Analysis of hysteretic curve characteristics under different confining pressures.

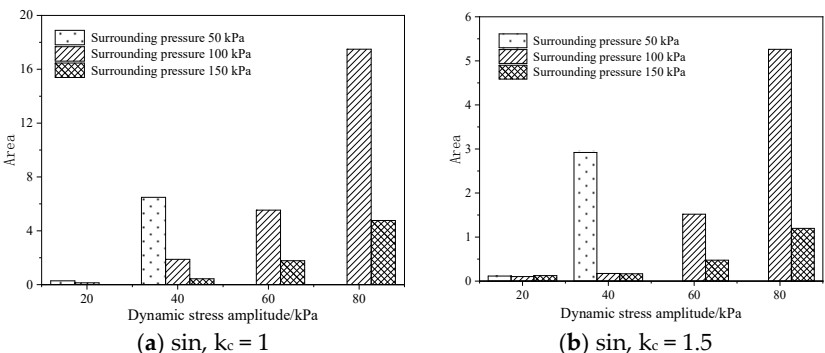

**Figure 7.** Effect of confining pressure on the surrounding area of hysteretic curve.

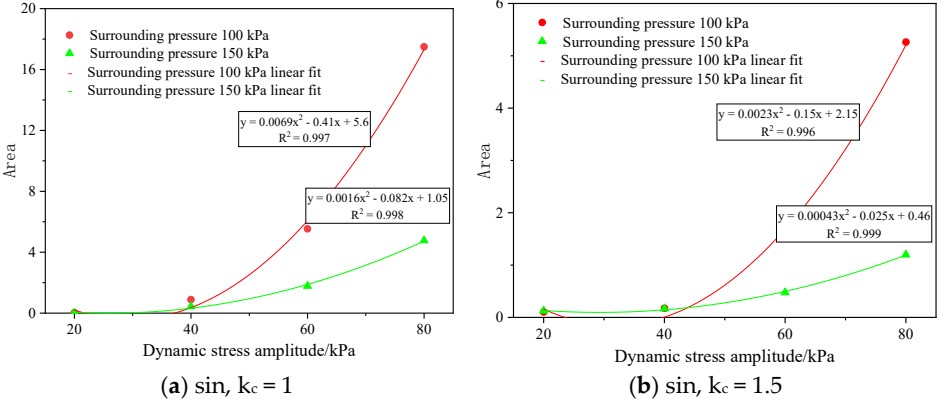

**Figure 8.** The second-degree parabola fitting of enclosed area for hysteretic curve under different confining pressures.

### 3.2. Analysis of Hysteretic Curve Characteristics under Different Consolidation Ratios

Figure 9a–f shows the influence law of different consolidation ratios on the hysteretic curve of the coarse-grained soil samples. The hysteretic curve generally had similar laws. When the dynamic stress amplitude was small, the hysteretic curve was compact, the slope was large and its shape was closer to an ellipse. With the gradual increase in the dynamic stress amplitude, the corresponding dynamic strain also increased gradually, the curve was elongated and the slope decreased. Under the same amplitude, the hysteretic curve became increasingly loose with the increase in cycle times.

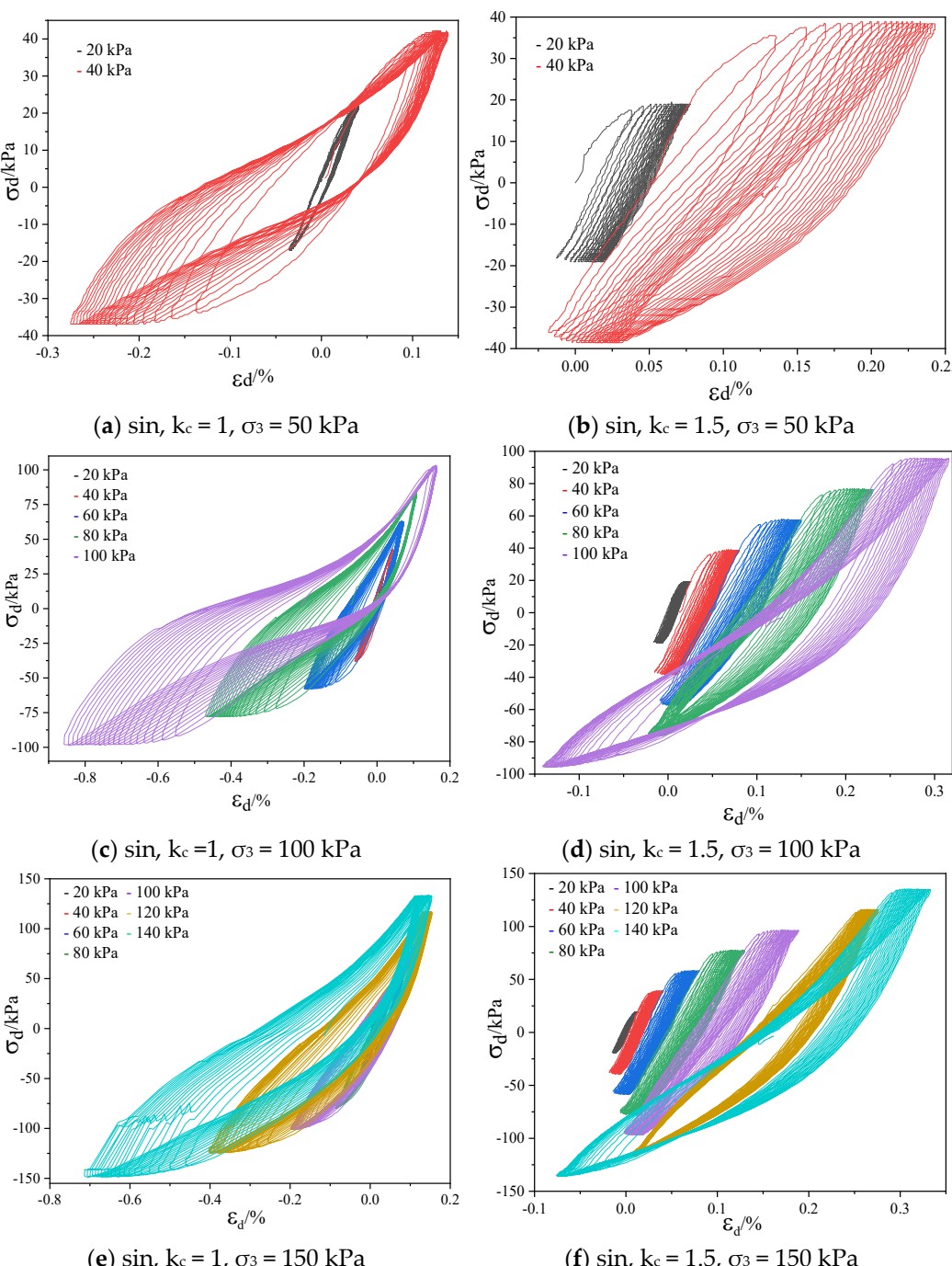

**Figure 9.** Effect of consolidation ratio on hysteretic curve.

Under the same dynamic stress level, the strain of the specimen response under bias consolidation was small. To be specific, in the hysteretic curve under the confining pressure

of 100 kPa, when the dynamic stress amplitude was 60 kPa, the strain amplitude of the isobaric consolidation was 0.175%, and the strain amplitude of the bias consolidation was 0.138%. When the dynamic stress amplitude was 80 kPa, the strain amplitude of the isobaric consolidation was 0.47%, and that of the eccentric consolidation was 0.231%. When the dynamic stress amplitude was 100 kPa, the strain amplitude of the isobaric consolidation was 0.858%, and that of the anisotropic consolidation was 0.316%, showing that the method of bias consolidation inhibited strain development of the coarse-grained soil specimens relatively well with the increasing stress levels.

We selected the hysteresis curve of the tenth cycle under each stage of loading and calculated the area surrounded, as shown in Figure 10. This analyzes the area variation characteristics enclosed by the hysteretic curve of the coarse-grained soil specimens under different consolidation ratios. Compared with isobaric consolidation, the area surrounded by the hysteretic circle was reduced by way of the bias consolidation. At a confining pressure of 100 kPa, its area decreased by 22%, 91%, 73% and 70%, respectively; at the confining pressure of 150 kPa, its area decreased by 90%, 61%, 73% and 74%, respectively. Increasing the consolidation ratio of the specimen was also conducive to reducing energy loss during the process of cyclic loading.

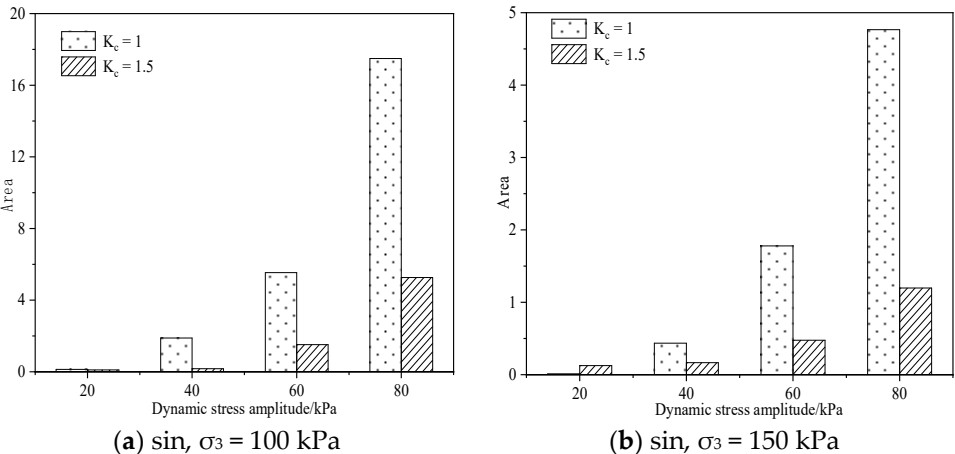

**Figure 10.** Effect of consolidation ratio on surrounding area of hysteretic curve.

The area surrounded by the hysteretic curve of the specimen under different consolidation ratios was the fitted second-degree parabola, as shown in Figure 11. The correlation coefficients of the linear fitting were all 0.99 and above. Under the same dynamic stress amplitude, the slope of the parabola tangent fitting for the eccentric consolidation was lower than that for the isobaric consolidation. It showed that under the same stress level, increasing the consolidation ratio can effectively reduce the area surrounded by the hysteretic curve. Increasing the consolidation ratio can effectively inhibit the increasing rate of its area with an increase in stress. This could control the energy consumption under a high stress level, showing better overall stability.

### 3.3. Analysis of Hysteretic Curve Characteristics under Different Cycle Times

Figure 12a–f shows the hysteretic curve under the test conditions of a confining pressure of 150 kPa and isobaric consolidation. The numbers chosen for the cyclic loading are: 1, 5, 10 and 15.

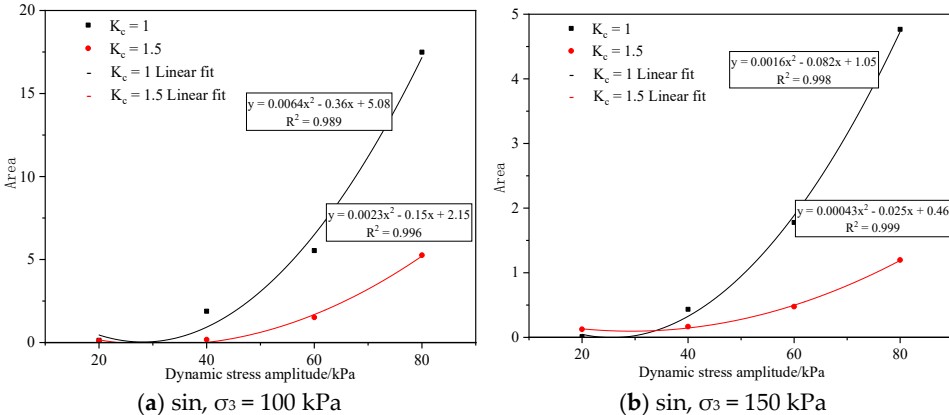

**Figure 11.** The second-degree parabola fitting of surrounding area of hysteretic curve under different consolidation ratio.

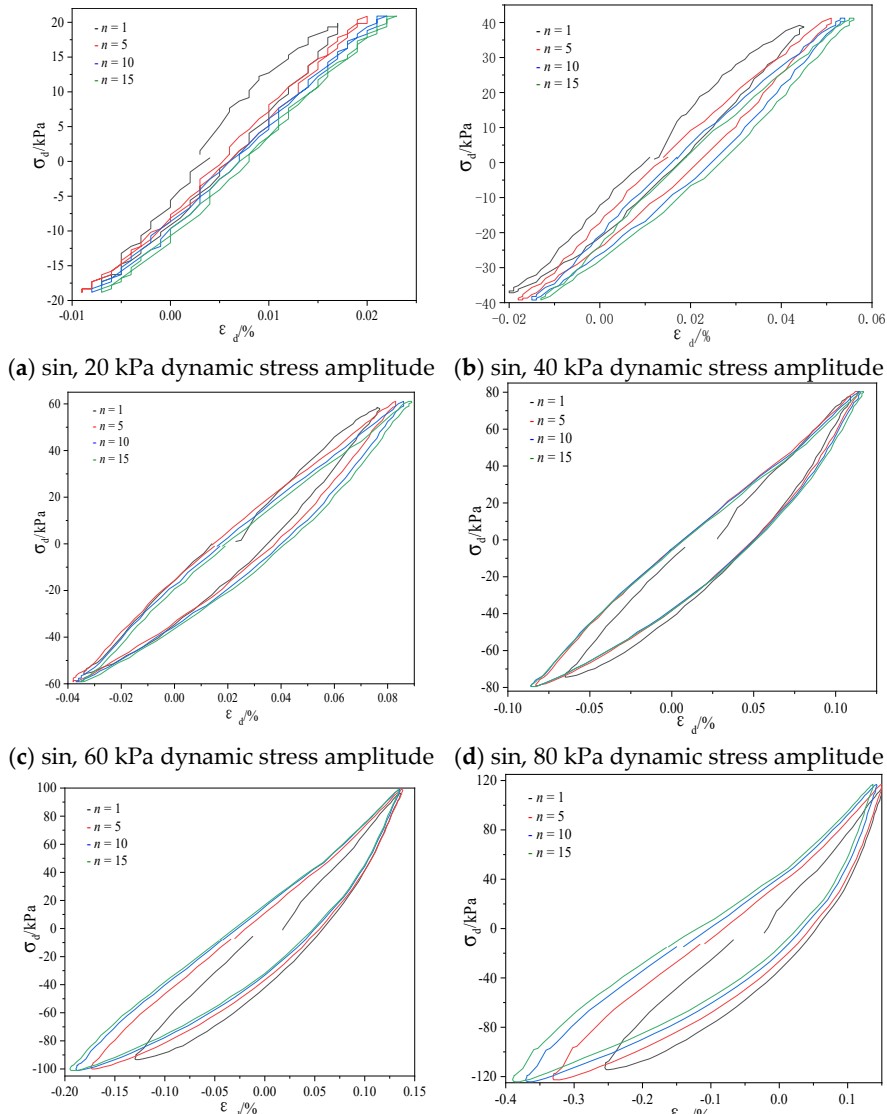

**Figure 12.** Characteristic analysis of hysteretic curve under different cycle times.

It can be seen that the number of cycles has little effect on the hysteretic curve at a low dynamic stress amplitude. The hysteresis curve is narrow. Under different cycle times, each hysteretic curve basically coincided. The amplitude of the strain, the area surrounded by the hysteresis curve and the slope performed little changes. When the dynamic stress amplitude reached 120 kPa, the hysteretic curve began to deviate greatly with the increase in cyclic loading times. In the case of a small dynamic stress amplitude, within a certain number of cycles, the hysteretic curve of the coarse-grained soil specimen was less affected by the number of cycles. In the case of a large dynamic stress amplitude, the hysteretic curve of the coarse-grained soil specimen was greatly affected by the number of cycles. Therefore, it can be concluded that when the stress reaches a certain level, the response strain changes greatly with the increase in cyclic loading times.

## 4. Conclusions

In this paper, the GDS dynamic triaxial apparatus, imported from Britain, was used to study the effects of the confining pressure, consolidation ratio and cycle action times on the hysteretic curve of coarse-grained soil; some achievements are summarized as follows:

(1) Increasing the confining pressure can densify the internal spatial structure of soil. Under the same stress level, an increase in the confining pressure reduced the strain of the specimen, which shows that increasing the confining pressure can effectively inhibit the development of soil deformation. Increasing the confining pressure can significantly reduce the energy consumption of the coarse-grained soil specimen under cyclic load. The decrease in energy consumption shows that the stiffness of the coarse-grained soil specimen improved; the overall stability of the specimen improved, and it shows better strength characteristics.

(2) Compared with isobaric consolidation, biased consolidation can inhibit the strain development of coarse-grained soil specimens better with an increasing stress level. Under the same stress level, increasing the consolidation ratio can effectively reduce the area surrounded by the hysteretic curve. Increasing the consolidation ratio can effectively inhibit the increase rate of its area with the increase in stress, thus controlling energy consumption under a high stress level, so as to show better overall stability.

(3) In the case of a small dynamic stress amplitude, the hysteretic curve of the coarse-grained soil specimen was less affected by the number of cycles within a certain number of cycles. In the case of a large dynamic stress amplitude, the hysteretic curve of the coarse-grained soil specimen was greatly affected by the number of cycles. When the stress reaches a certain level, the response strain changes greatly with the increase of cyclic loading times.

The purpose of this paper is to solve the problem of dynamic deformation caused by road traffic load and intuitively explore the strain development law of coarse-grained soil under a dynamic load. Later, through numerical simulation analysis and field stress-strain monitoring, we compare the test data to obtain better conclusions and methods for solving practical problems. In the future, we will analyze the backbone curve, dynamic modulus and damping ratio of coarse-grained soil. Further, we will try to use IRI performance models [21] and Fuzzy logic [22] to predict subgrade dynamic performance.

**Author Contributions:** Data curation, W.H. and Z.Y.; Investigation, P.L. and W.Y.; Methodology, P.W. and Q.Z.; Supervision, X.K.; Writing—original draft, P.W. All authors have read and agreed to the published version of the manuscript.

**Funding:** This research was funded by Shandong Provincial Natural Science Foundation (Grant No. ZR2021ME238), Science and Technology Project of Universities in Shandong Province (Grant No. J18KA216), and Science and Technology Project of Universities in Shandong Province (Grant No. J18KA192).

**Institutional Review Board Statement:** Not applicable.

**Informed Consent Statement:** Not applicable.

**Data Availability Statement:** The authors confirm that the data supporting the findings of this study are available within the article.

**Conflicts of Interest:** The authors declare no conflict of interest.

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
