# Peer review of "An Experimental Study on Dynamic Characteristics of Coarse-Grained Soil under Step Cyclic Loading"

_coatings, doi:10.3390/coatings12050640_

Round 1

Reviewer 1 Report

The paper can be published after the corrections have been made. Please send the paper again after the authors make corrections.

Reviewer 2 Report

This is a well-written and carefully structured paper that presents a characterization of the in-situ response and performance of road structures.

I have a few comments that might be usefully addressed to improve the overall quality of the paper:

  • The experimental program is detailed and the quality of the results’ discussion in its current form satisfies the requirements necessary for a research paper.
  • The problem is at the conclusion, because the results obtained are logical results, considering the classic connection between the loading effort and the lateral pressure. The article must highlight the way of putting into practice the results obtained so as to result in a set of conclusions corresponding to the phenomenon studied in the laboratory. Or, if the study stops only in the laboratory part, the conclusions are insufficient, they present only a normal course of the classical effort-deformation link.

It is mandatory to review the test program and the related conclusions.

The paper is therefore very well suited to this journal. The authors are to be commended on the professional quality of the research and the paper. Also, some thoughts are due on future developments, especially on the practical applicability of this method.

Reviewer 3 Report

The manuscript presents the study of coarse-grained soil fillers evaluated by the GDS dynamic triaxial test with the aim of evaluating the influence of different circumferential pressure, consolidation ratio and number of cyclic actions of the hysteresis curve under step cyclic loading.

The results and the discussion are   conveniently presented. Nonetheless, the “introduction” and the “test preparation” sections are not so adequately presented.

Firstly, when presenting that the coarse-grained soils of the subgrades in the pavements are affected by the traffic load, it should be convenient to introduce the pavement structure, surface layer, base, and subbase, over a compacted subgrade, as in Perez-Acebo et al. 2020 and in Olowosulu et al. (2021), for example. In fact, the subgrade is where the materials that are tested in this paper are located and it is not indicated which type of pavements is for: flexible, semi-rigid or rigid. A figure of the usual pavement structure is also convenient. Additionally, the way that loads arrive to the subgrade may be included.

Secondly, when in line 59 it is said that the soil sample was taken from a typical coarse-grained soil along a new highway, more information should be introduced: where it was taken from. Why is it a typical soil? According to which standard? Perhaps, the employment of typical is not adequate. In other locations, other types of soil would be typical.

Line 63 should be rewritten: “The mass percentage of soil particles with a diameter above 0.075 mm (d ≥ 0.075 mm) in the coarse-grained group of soil samples was 91.7 %...” In the final part of the same sentence, “and was coarse-grained soil [11]”, it should be better to say that “(the sample soil) was classified as a coarse-grained soil, according to Chinese standard JTG 3430-2020 [11]”. Moreover, another soil classification should be convenient, as the ASTM. Thus, more readers could understand what kind of soil it is.

Furthermore, the paragraph 82-86 is perhaps the one to be improved. It is said that four aspects of the sample parameters were selected: soil conditions, drainage conditions, stress conditions and dynamic conditions. How were they measured? It must be explained which are exactly the variables that are employed, obtained by which test, and the range of the values of each variable. Similarly, it is said that the main influencing factors chosen were confining pressure, dynamic stress amplitude, loading frequency and the number of cycles. How are those values measured? Explanations are needed. Additionally, perhaps, a flow diagram is needed to understand the entire researching process.

Moreover, in Figure 8a, it is difficult to believe that the red line has a R2= 0.80, and the black line, whose points are more distant to the line, has obtained a R2 of 0.85. The same happens in Figure 8b. R2 = 0.88 seems very high according to the observed distance from the points to the line. Perhaps, a second degree parabola would be more adequate. Similar problems are observed in Figure 5. Very high R2 are obtained.

Finally, with regard to the form of the article, authors did not respect some of the points of the journal’s template. References are introduced in the text as number between squared parentheses, but not placed as super indices. Additionally, references at the final part of the article did not respect the template. Author’s name must not be in capital letter, journals’ name must be abbreviated, etc. Please, follow journal’s template.

Minor points:

GDS is included in the abstract and in line 54 but it is not defined.

Figure 5 must be placed after it is mentioned, not before.

Perez-Acebo et al. (2020). IRI performance models for flexible pavements in two-lane roads until first maintenance and/or rehabilitation work. Coatings, 10 (2), 97.

Olowosulu et al. (2021). Development of framework for performance prediction of flexible road pavement in Nigeria using Fuzzy logic theory. International Journal of Pavement Engineering, 10.1080/10298436.2021.1922907

Round 2

Reviewer 1 Report

Suggestions are duly accepted, I have no further objections.

Author Response

Thanks

Reviewer 2 Report

The paper can be accepted in present form.

Author Response

Thanks.

Reviewer 3 Report

The paper has been considerably improved. However, I will insist in some points:

A figure showing the layers of a pavement structure would be adequate, to clearly indicate where the subgrade is, as the one shown in Olowosulu et al. (2021). The best place would be after line 59.

I would remove the word “typical” from line 73. There is no typical subgrade as the geology changes in each place.

Line 87. I would include the description of those coefficients as equations.

Author Response

Points 1: A figure showing the layers of a pavement structure would be adequate, to clearly indicate where the subgrade is, as the one shown in Olowosulu et al. (2021). The best place would be after line 59.

Response:Thanks a lot. After reading the two references (IRI Performance Models for Flexible Pavements in Two-Lane Roads until First Maintenance and/or Rehabilitation Work. Development of framework for performance prediction of flexible road pavement in Nigeria using Fuzzy logic theory. International Journal of Pavement Engineering.) carefully, we found that they would help us to predict subgrade dynamic performance, and a figure was not necessary for us. We have quoted the two references in the end of my article, which is more appropriate.

Points 2: I would remove the word “typical” from line 73. There is no typical subgrade as the geology changes in each place.

Response:Thanks for your suggestion. I have deleted the word “typical”.

Points 3: Line 87, I would include the description of those coefficients as equations.

Response:Thanks for your suggestion. I have included the description of those coefficients as equations.